# Fenestrated Physician-Modified Endografts for Preservation of Main and Accessory Renal Arteries in Juxtarenal Aortic Aneurysms

**DOI:** 10.3390/jcm12144708

**Published:** 2023-07-15

**Authors:** Hon-Lai Chan, Dimitrios D. Papazoglou, Silvan Jungi, Salome Weiss, Daniel Becker, Drosos Kotelis, Vladimir Makaloski

**Affiliations:** Department of Vascular Surgery, Inselspital, Bern University Hospital, University of Bern, 3010 Bern, Switzerland; chanhonlai@yahoo.fr (H.-L.C.); vladimir.makaloski@insel.ch (V.M.)

**Keywords:** endograft, physician-modified, juxtarenal, fenestration, endovascular aortic repair

## Abstract

Background: There is a paucity of reporting outcomes of complex aortic aneurysm treatment such as juxtarenal abdominal aortic aneurysms, where additional techniques to preserve renal artery perfusion are required. Methods: Retrospective analysis of consecutive patients who underwent emergent and elective aortic repair with fenestrated PMEGs between March 2019 and January 2023. Endpoints were technical success, reinterventions, secondary reinterventions and target vessel patency. Results: Forty-seven target vessels in 37 patients (23 male, median age 75 years) were targeted, of which 44 were renal arteries (RAs) with a mean diameter of 5.4 ± 1.0 mm. Thirteen were accessory RAs and six had a diameter ≤ 4 mm. Technical success rate was 87% overall; 97% for main and 62% for accessory RAs respectively. Target vessel patency and freedom from secondary reintervention was 100% and 97% at 30 days and 96% and 91% at one year, respectively. There was no 30-day mortality. Conclusion: Fenestrated physician-modified endografts are safe and effective for the treatment of patients with juxtarenal abdominal aortic aneurysms when incorporating main renal arteries. Limited technical success may be expected when targeting accessory renal arteries, especially when small in diameter. Long-term follow-up is needed to confirm durability of PMEGs for renal artery preservation.

## 1. Introduction

Endovascular aortic repair has dramatically changed the landscape of aortic aneurysm treatment with a reduction in operative morbidity and mortality when compared with open repair [1,2,3]. This especially applies to juxtarenal abdominal aortic aneurysms (AAAs), which would require suprarenal aortic cross-clamping [4,5,6]. During endovascular repair, juxtarenal AAAs require additional endovascular techniques to preserve renal artery perfusion while achieving an adequate proximal sealing zone for the endograft. The same techniques may become necessary to preserve accessory renal arteries (ARA), which are present unilaterally in 25% and bilaterally in 10% of the population [7]. ARA coverage may lead to renal infarction, potentially resulting in long-term renal insufficiency [8,9]. Therefore, preservation of ARAs with a diameter ≥ 4 mm is recommended [10].

Fenestrated endovascular aortic repair (FEVAR) is a well-established, safe and durable treatment option for juxtarenal AAAs [11,12]. The production of these custom-made devices may take up to 16 weeks, generating relevant costs and potentially lethal treatment delays [13,14,15]. In contrast, the use of off-the-shelf fenestrated or branched devices, readily available in emergencies, may be limited by patients’ anatomy and concerns regarding long-term branch stability in the setting of misaligned fenestration/branches and bridging stents have been raised [16,17,18]. Furthermore, off-the-shelf devices preclude the incorporation of ARAs and may result in additional aortic coverage with sometimes unnecessary inclusion of the visceral vessels, which is associated with increased morbidity and mortality [19]. Other techniques used for renal artery preservation in juxtarenal AAAs, such as the chimney, periscope or sandwich technique, have mostly been abandoned due to low target vessel patency and high rates of endoleaks [20,21,22,23,24,25]. Fenestrated physician-modified endografts (PMEGs) are created by modification of conventional, off-the-shelf available endografts. This technique has been developed to overcome the limits of custom-made and off-the-shelf fenestrated/branched devices and good results have been shown in the elective and emergency setting by experienced centers [26,27,28,29,30,31,32]. However, their long-term durability is still under-studied and little information exists on their use for the preservation of main and accessory renal arteries specifically. The purpose of this report is to review our experience with PMEGs for the treatment of juxtarenal aortic aneurysms with the preservation of main or accessory renal arteries.

## 2. Methods

Consecutive patients who underwent emergent and elective PMEG aortic repair for juxtarenal AAAs at a single institution between March 2019 and January 2023 were analyzed. The off-label nature of the PMEG was disclosed in all cases, and written informed consent for the intervention and the further use of their health-related data and images was obtained from all patients. The local ethics committee waived the need for approval due to the low number of patients. All data were extracted from medical records and available imaging studies. Life status was assured on the study reporting date. Follow-up index was calculated as previously suggested [33]. Collected data are shown in Appendix A. Juxtarenal AAA was defined according to the current guidelines as an aneurysm extending up, but not involving, the renal arteries with a short infrarenal neck < 10 mm [34]. This report adheres to the Society for Vascular Surgery reporting standards for endovascular aortic aneurysm repair and endovascular repair of aneurysms involving the renal-mesenteric arteries [35,36]. Statistical analysis was performed using Stata version 15 (StataCorp LLC, College Station, TX, USA).

The indication for repair of a juxtarenal AAA was based on a patient’s general condition, comorbidities and patient preference and the decision to use a PMEG was based on aneurysm and renovisceral anatomy. The decision-making process for treatment of juxtarenal AAAs in our department is explained in Appendix A. For this study, comorbidities were retrospectively graded using the Society for Vascular Surgery (SVS)/American Association for Vascular Surgery (AAVS) medical comorbidity grading system [37]. Preoperative CT angiography (CTA) was available for all patients for preoperative planning and device modification. Postoperatively, follow-up CTAs were performed at standard intervals one, six, and twelve months after the procedure (and yearly thereafter). Acute kidney injury was defined according to the RIFLE/KDIGO criteria and long-term renal function impairment was defined as a drop of ≥ 20% of glomerular filtration rate (GFR) compared with values at admission. For this study, all available imaging studies were reviewed by the first authors using multiplanar reconstruction (SECTRA PACS, Sectra AB, Linköping, Sweden) and centerline reconstruction in a vascular imaging workstation (OsiriX MD, 64-bit; Pixmeo, Geneva, Switzerland). The infrarenal neck length was defined as the segment of the aorta inferior to the lowest renal or accessory renal artery with parallel aortic wall with minimal (<10%) or no change in diameter and no atherosclerotic debris, thrombus or calcification. The length between the most distal point of the lowest visceral branch not included in the fenestrated PMEG and the beginning of the aneurysm was set as the proximal landing zone and the difference between the proximal landing zone and infrarenal neck length was termed the gained landing zone through fenestrated PMEG.

### 2.1. Physician-Modified Endograft Technique

The Endurant II stent graft system (Medtronic, Minneapolis, MN, USA) was used to create the fenestrated PMEGs. Incorporating the necessary renal arteries, we aimed for a proximal sealing zone of a healthy, parallel aortic wall with a minimum length of 10 mm, as recommended in the instructions for use for standard infrarenal deployment of the Endurant II endograft, but preferred a length of 15 mm wherever possible. The standard length used for aortic cuffs was 49 mm, and 145 mm or 166 mm for bifurcated endografts. We aimed for a 10% oversizing in aortic cuffs and 15–20% oversizing for bifurcated endografts. Our physician-modified fenestration technique has already been described previously (Figure 1) [38]. In brief, the endograft was partially deployed back-table under strict sterile conditions. The fenestration was created with a scalpel and reinforced with a wire from a snare catheter (En Snare, Merit Medical Systems, South Jordan, UT, USA) and two layers of braided 5-0 non-absorbable sutures. The modified endograft was re-loaded into the delivery system using vascular tourniquets. For implantation, bilateral percutaneous common femoral artery access was obtained. After endograft deployment, renal artery catheterization was usually achieved from a femoral access, but additional left upper extremity access was established when necessary. Bridging stents were implanted through the fenestrations into the target vessels using balloon-expandable covered stents. Bridging stents equal in size to—or 1 mm larger than—the target renal artery diameter were selected. The Advanta V12 bridging stents (Atrium Maquet, Hudson, NH, USA) were used in case of a target renal artery diameter of ≥5 mm and PK Papyrus-covered coronary stent grafts (Biotronik SE & Co. KG, Berlin, Germany) were used for diameters of <5 mm. All bridging stents were additionally flared at the fenestration with a balloon that was at least 2 mm larger, to ensure adequate sealing within the main graft. Patients were under heparinization (activated clotting time ≥250 s) and procedures were performed using fusion imaging in a hybrid operating room with a fixed imaging system (Allura Clarity, Philips, Best, The Netherlands), either under local anesthesia with anesthesiologic surveillance or under general anesthesia.

### 2.2. Endpoints and Definitions

Endpoints were technical success, reinterventions, secondary reinterventions and target vessel patency, as proposed by the Society for Vascular Surgery reporting standards [36]. Technical success was defined as successful access to the arterial system, deployment of the aortic stent graft and all modular stent graft components, successful catheterization and placement of bridging stents with maintenance of flow in all intended target vessels, the absence of type I or type III endoleaks and patency of all aortic modular stent graft components at completion angiography [37]. Reinterventions were defined as major procedures designed to treat the underlying aortic disease such as open conversion, endovascular or open intervention for endoleaks. Secondary reinterventions included treatment of branch vessel stenosis or occlusion, leg stenosis or occlusion or embolization, as previously proposed [36]. The patient with open conversion was excluded from target vessel patency and secondary reinterventions analyses due to different treatment, but included in all other analyses.

## 3. Results

During the study period, a total of 148 patients with juxtarenal AAAs were treated in our institution, of which 37 patients (25%) with a median age of 75 years (range 61–89) were treated with PMEG (Appendix A). All preoperative characteristics are listed in Table 1.

Thirty patients were treated for degenerative juxtarenal AAAs, five were treated for a type Ia endoleak after previous EVAR and two patients were treated due to a proximal anastomotic pseudoaneurysm after open infrarenal AAA repair without evidence of infection. Three interventions were emergent or urgent (two ruptures, one symptomatic aneurysm), and in 34 patients the intervention was in an elective setting. Risk class of the American Society of Anesthesiology (ASA) was ≥ 4 in 43% and median Society for Vascular Surgery total score was 6 (range 1–13) (Table 1). [37]. Mean aneurysm diameter was 64.0 ± 17.8 mm and mean proximal landing zone length was 26.9 ± 10.7 mm. Forty-seven target vessels were targeted, of which 44 were renal arteries with a mean diameter of 5.4 ± 1.0 mm. Thirteen were ARAs and 6 had a diameter ≤ 4 mm (Table 2).

### 3.1. Perioperative Outcomes

Mean operation time including the physician modification of the endograft was 166 min (range 72–300). In the three urgent/emergent patients the procedure lasted 105 min for a single fenestrated physician-modified aortic cuff for contained rupture after EVAR due to type Ia endoleak, 180 min for a threefold fenestrated physician-modified bifurcated graft in case of contained rupture and 195 min for a threefold physician-modified fenestrated bifurcated graft for an 11 cm symptomatic aneurysm. Technical success was 87% overall; 100% for the superior mesenteric artery, 97% for the main renal and 62% for ARAs (Table 3). One main renal artery could not be catheterized during initial intervention, but was successfully bridged the day after without any change in renal function. Revascularization of five ARAs in five patients was not possible (median diameter 4 mm, range 3–5 mm), due to three catheterization failures, one dislocation of the bridging stent graft during intervention, and one misalignment of the fenestration in regard to the target vessel. The misalignment of the fenestration was due to a partially infolded main graft and resulted in a type III endoleak with no option for endovascular bailout. Therefore, an open conversion with suprarenal clamping and partial replacement of the main body as well as re-implantation of the ARA followed two days later. At completion angiography there were five type II and two type III endoleaks (one in the patient with failed main renal artery cannulation at the primary intervention and one in the patient with failed ARA revascularization due to graft infolding). In the remaining four patients with non-revascularized ARAs, no endoleak was detected at discharge or at follow-up (Appendix A).

Reintervention rate was 5% (2/37) as mentioned before. There were no neurological, cardiac, respiratory or gastrointestinal complications and no deaths within 30 days. The 30-day target vessel patency was 100% (Figure 2). One patient with intraabdominal compartment syndrome treated for a ruptured AAA needed decompressive laparotomy. One patient had an infected lymph fistula in the groin, which resolved after multiple revisions and negative-pressure wound therapy. Further outcomes are shown in Table 3. All ARA revascularization failures occurred in the first half of the patients. Acute kidney injury occurred in two patients during hospitalization: one with the open conversion and suprarenal clamping and the other one with bridging of the main renal artery the day after initial intervention. Both patients reached their preoperative renal function parameters prior to discharge without any need for dialysis. There was no difference between the renal function at admission and discharge for creatinine (97 vs. 101 umol/L, *p* = 0.43) and GFR (68 vs. 67 mL/min, *p* = 0.80), respectively.

### 3.2. Follow-Up

Median follow-up was 17.7 months (range 1–47) and all patients had a CTA at discharge or at follow-up. Follow-up was complete for all patients until the study closure date and follow-up index was 1.0. Freedom from secondary reinterventions was 91% at one year (Figure 3). Four patients underwent eight secondary reinterventions, which were all due to graft limb occlusions. Two patients with kinked stent graft limbs were successfully relined and no sequential occlusion occurred. One patient embolized from a dilated common iliac artery, which was excluded with a physician-modified iliac branch device and in one patient a hematologic disorder with increased thrombogenicity was suspected as the cause, without any signs of kinking or stenosis of the stent graft. All patients had restoration of below-the-knee vessel perfusion and no amputation was needed. No interventions were necessary for the physician-modified fenestrations or the bridging stents during follow-up and there was no renal function impairment of >20% of GFR during follow-up.

Target vessel patency was 96% at one year (Figure 2). The ARA with the dislocated bridging stent occluded during follow-up with subsequent small renal infarction without a worsening of renal function. All other target vessels as well as the remaining four non-stented ARAs remained open (Appendix A).

Twelve (33%) type II endoleaks were detected during follow-up (Table 4). Decreased or stable aneurysm diameter was measured in 36 (97%) patients and increased aneurysm diameter of ≥5 mm compared with the preoperative diameter in one patient. Four (11%) patients died during follow-up, all from non-aneurysm related causes and not related to the procedure or device (Figure 4). One-year overall survival was 93%.

## 4. Discussion

Juxtarenal AAAs pose a challenge to endovascular aortic repair. Fenestrated custom-made devices have been proven to be safe and effective in the elective setting but there are limitations in regards to production time as well as costs [5,11,13,15]. PMEGs have shown excellent results in experienced centers in the emergency as well as the elective setting [27,32]. We present our early results of PMEGs in juxtarenal AAAs with preservation of main or accessory renal arteries.

The most important complication in our series was one early open conversion due to malposition of a partially infolded main body, which precluded renal artery cannulation, resulting in a type III endoleak with no further endovascular treatment option. In this patient, the main body used for physician modification had a diameter of 36 mm and re-sheathing into the 20 French delivery system was very demanding. We assume that the infolding happened during re-sheathing. Therefore, this could probably have been avoided, if a properly loaded, custom-made device had been used for this patient. However, we had no issues while re-sheathing the main bodies in the other patients. It remains unclear whether this was a single event or if this is generally an issue when re-sheathing larger stent grafts, but physicians performing these procedures should be aware of it.

Our 87% overall technical success rate was lower than in other reports [27,28,29,30]. However, for main renal arteries only, primary technical success was 97% and secondary technical success was 100% after bridging one renal artery the day after the initial procedure. This patient had a severely calcified, angulated and narrow aortic neck, which interfered with target vessel catheterization. Thus, technical failure concerned mainly ARAs, of which five could not be targeted with a bridging stent during the index procedure. Two of them were 3 mm and one of them was 4 mm in diameter, which is known to bear a high risk of technical failure [10,39]. Although two reports have shown that coverage of small ARAs with a median diameters of 3 and 3.4 mm (range 2.5–4 mm) were associated with renal function deterioration after 6 and 12 months, respectively, it is recommended to preserve only ARAs with a diameter of ≥ 4 mm due to high technical failure rates, risk of renal artery disruption and kidney loss [10,34,39,40,41]. In our series, ARAs below a diameter of 4 mm were selected as target vessels when preoperative CTA suggested that they supplied at least one-fourth of the renal parenchyma. Interestingly, in our series, catheterization failure did not lead to type III endoleak, except in the patient with partial infolding of the main body and consecutive conversion. In these patients, the ARAs branched off in the proximal aortic neck, which was completely sealed by the endograft, preventing any endoleak. Furthermore, four of the five not-connected ARAs remained patent and we observed an overall target vessel patency of 100% at 30-days and 96% at one year. Misalignment was the cause in one catheterization failure which we tried to prevent by leaving the tip capture closed until one target vessel is cannulated and align the fenestration with the target vessel with a balloon maneuver.

Our 30-day and one-year freedom from secondary reintervention rates of 100% and 93% are slightly better compared with reports from custom-made fenestrated devices [42,43]. Off-the-shelf devices have shown to be prone to kinking and stenosis of target vessels, which resulted in higher secondary reintervention rates in recent reports [44,45]. This may be explained by the fact that, in these reports, more patients were treated by emergency, and both juxtarenal and thoracoabdominal aortic aneurysms were included. Additionally, we selected our patients with a favorable anatomy for fenestrated PMEG (at least those in the elective setting), possibly leading to the decreased need for secondary reinterventions during follow-up [26,27,32,46].

In our population with juxtarenal AAAs, the landing zone was extended through the endograft modification by a mean of 18.9 ± 11.4 mm to a total mean landing zone of length of 26.9 ± 10.7 mm (Table 2). This is clearly a shorter landing zone than in other studies on juxtarenal AAAs treated with FEVAR and including additional visceral branches and extended aortic coverage [46,47]. Nevertheless, no type Ia endoleaks occurred in our series. Inclusion of more reno-visceral branches and extension of the proximal landing zone have been associated with increased peri-operative mortality and morbidity [19,48]. However, as the aneurysmal disease progresses, shorter proximal landing zones could be prone to type Ia endoleaks during long-term follow-up. Our very low rate of type I and III endoleak and favorable aneurysm sac development are comparable with recent reports [32,44]. As observed by Oderich et. al. significant differences in aneurysmal sac changes and freedom from secondary reinterventions between physician-modified and custom-made fenestrated-branched stent grafts appear after a follow-up of one year [32]. Therefore longer follow-up is needed to verify our results and long-term durability of fenestrated PMEGs for the preservation of main and accessory renal arteries in juxtarenal AAAs.

Although custom-made devices, due to standard production and rigorous testing, should be first-line choice in the elective setting, fenestrated PMEGs have shown excellent results if implanted by experienced physicians in selective cases [27,32]. These results are partly better than those with off-the-shelf devices, which may be due to the patient-specific design of the PMEG [4]. Multiple reports have proven that results of fenestrated PMEGs improve significantly with expertise, which is also true in our experience [26,32]. In addition, 3D-printed aortic models are a promising tool for easier modification of endografts by increasing the accuracy of the fenestrations and therefore possibly enabling their broader use [49,50,51].

Custom-made devices have a production and delivery time of up to 16 weeks, during which a substantial proportion of patients experience aneurysm rupture [15]. Additionally, a custom-made device costs approximately three times more than an off-the-shelf EVAR stent graft which is modified [13,14]. In times of rising economic health care burden these interventions are an important driver of health care costs. When performed in suitable elective cases, the physician-modification of endografts leads to an increased experience and better results for emergency cases as well.

### Limitations

This study has several limitations due to its retrospective nature and small sample size. Additionally, patients were highly selected for this treatment and therefore generalizability is very limited.

## 5. Conclusions

Patients with juxtarenal abdominal aortic aneurysms can safely and effectively be treated with fenestrated physician-modified endografts incorporating main and accessory renal arteries with good short-term results. Accessory renal arteries with a diameter of ≤4 mm bear a high risk of technical failure. Long-term follow-up is needed to confirm durability.

## Figures and Tables

**Figure 1 jcm-12-04708-f001:**
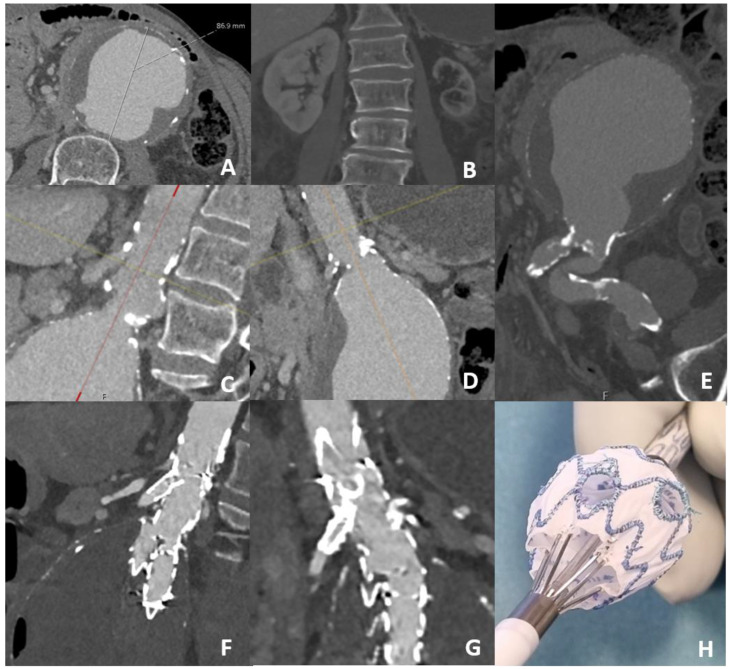
71-year-old-male with Ileo-colic resection and jejunostomy one month before presenting himself in the emergency room with: (**A**) a symptomatic 87 mm juxtarenal aortic aneurysm, (**B**) occluded left renal artery with shrunken kidney, (**C**,**D**) diffuse calcification of the visceral aorta including the origin of the left renal artery (side- and frontal-view), (**E**) with severely calcified and kinked left common iliac artery (**F**) side-view of the physician-modified endograft with the bridging stent to the superior mesenteric artery, (**G**) frontal-view with the bridging stent in the right renal artery, (**H**) twofold fenestrated physician-modified endograft after back-table modification.

**Figure 2 jcm-12-04708-f002:**
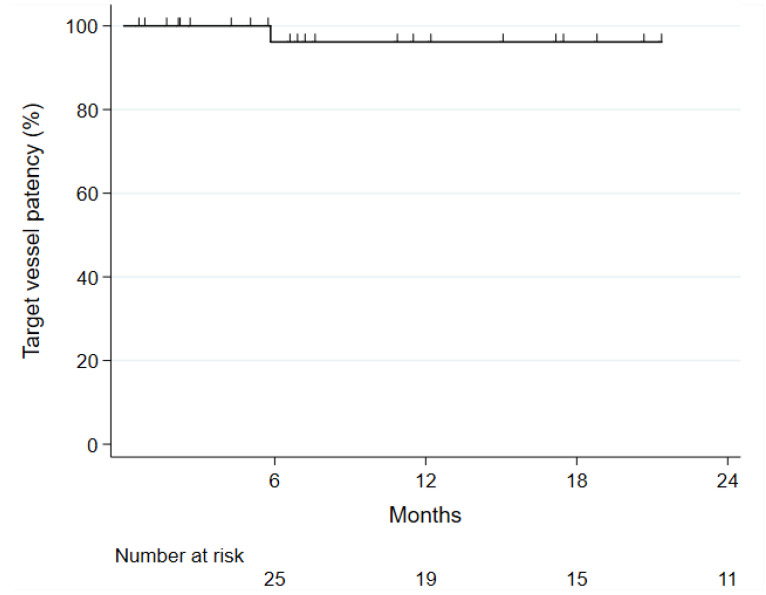
Kaplan–Meier curve of target vessel patency in 36 patients with juxtarenal aortic aneurysms treated with fenestrated physician-modified endografts.

**Figure 3 jcm-12-04708-f003:**
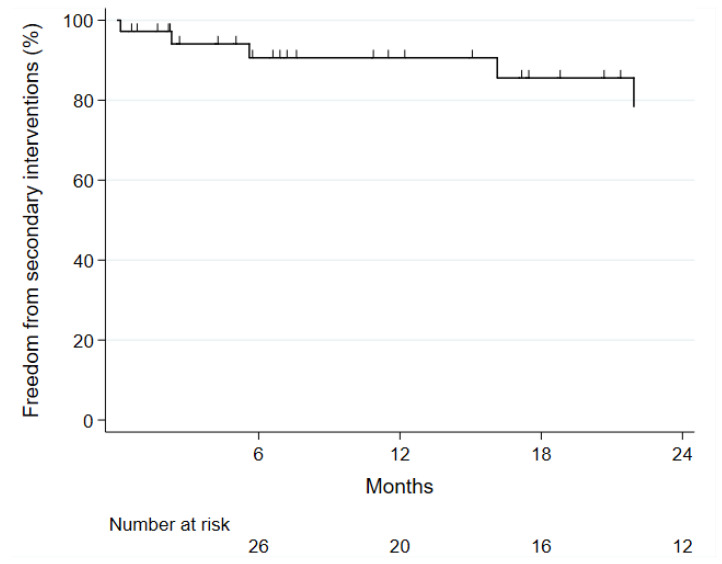
Kaplan–Meier curve of freedom from secondary reinterventions in 36 patients with juxtarenal aortic aneurysms treated with fenestrated physician-modified endografts.

**Figure 4 jcm-12-04708-f004:**
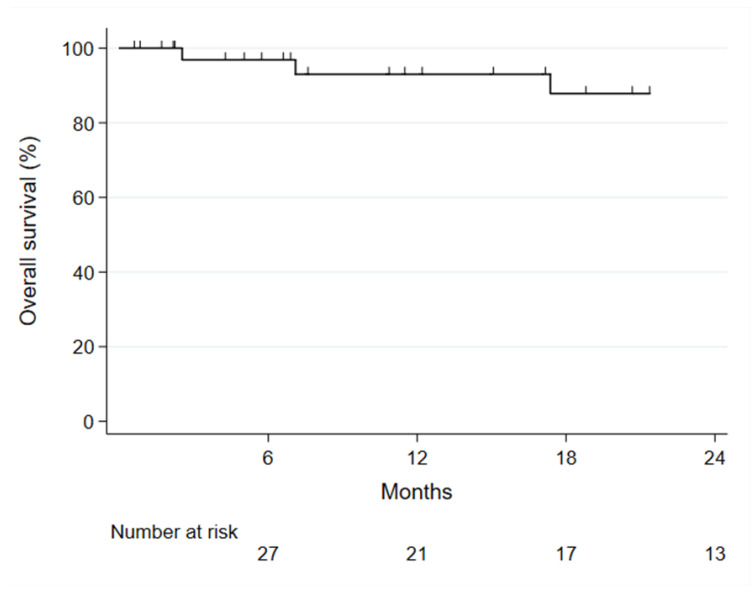
Kaplan–Meier curve of overall survival in 37 patients with juxtarenal aortic aneurysms treated with fenestrated physician-modified endografts.

**Table 1 jcm-12-04708-t001:** Demographics and clinical characteristics of 37 patients with juxtarenal aortic aneurysms undergoing endovascular repair with fenestrated physician-modified endografts.

All Patients (*n* = 37)	
Age (years), median (range)	75 (61–89)
Male	33 (89)
BMI (kg/m^2^), median (range)	25.8 (15–38)
Medical history	
CAD	13 (35)
Arrhythmia	6 (16)
Hypertension	32 (87)
Active smoker	14 (38)
COPD	9 (24)
Baseline creatinine (µmol/L)	97 (52–212)
Diabetes	6 (16)
Stroke/TIA	6 (16)
PAD	9 (24)
SVS score	6 (1–13)
ASA class ≥ 4	16 (43)
Previous medication	
Antiplateled therapy	33 (89)
Anticoagulation therapy	9 (24)
Statin	32 (87)
ACE inhibitor	27 (73)
Betablocker	15 (41)
Aortic history	
Previous aortic intervention	7 (19)
Endovascular AAA repair	5 (14)
Open AAA repair	2 (5)

Data are presented as n (%) unless specified otherwise. BMI = body mass index; CAD = coronary artery disease; TIA = transient ischemic attack; PAD = peripheral artery disease; COPD = chronic obstructive pulmonary disease; SVS = Society of Vascular Surgery; ASA = American Society of Anesthesiology; ACE = angiotensin converting enzyme; AAA = abdominal aortic aneurysm.

**Table 2 jcm-12-04708-t002:** Anatomical characteristics of 37 patients with juxtarenal aortic aneurysms undergoing endovascular repair with fenestrated physician-modified endografts.

All Patients (*n* = 37)	
Aneurysm diameter	64.0 ± 17.8
Infrarenal neck length	8.0 ± 5.5
Proximal landing zone ^a^	26.9 ± 10.7
Gained landing zone through pm-fenestrations ^b^	18.9 ± 11.4
Proximal cuff oversizing	11.4 ± 8.2
Proximal bifurcated graft oversizing	15.5 ± 8.0
Suprarenal angle (α)	18 ± 15°
Infrarenal angle (β)	33 ± 21°
Proximal neck thrombus/calcification > 50%	2 (5)
Iliac artery aneurysm	2 (5)
Iliac artery stenosis > 50%	4 (11)
Turtuos access vessels	0 (0)
Target vessels	47 (100)
One pm fenestration	29 (78)
Two pm fenestrations	6 (16)
Three pm fenestrations	2 (6)
Superior Mesenteric Artery	3 (6)
Renal arteries	31 (66)
Accessory renal arteries	13 (28)
Main renal artery size	5.8 ± 1.4
Accessory renal artery size	4.5 ± 1.5

Data are presented as n (%) and mean ± standard deviation [mm] unless specified otherwise. ^a^ Length between the most distal point of the lowest visceral branch not included in the fenestrated PMEG and the beginning of the aneurysm ^b^ Difference of length of proximal landing zone and infrarenal neck. pm = physician-modified.

**Table 3 jcm-12-04708-t003:** Peri- and postoperative outcomes of 37 patients with juxtarenal aortic aneurysms undergoing endovascular repair with fenestrated physician-modified endografts.

	All Patients (*n* = 37)
Total Operation Time, minutes	166 (72–300)
Volume of Contrast, mL	110 (41–480)
Fluoroscopy time, minutes	51 (13–106)
Dose area product, mGy/cm^2^	193,966 (23,001–624,789)
Blood loss, mL	394 (20–5000)
Technical success	
Superior mesenteric artery	3/3 (100)
Main renal arteries	30/31 (97)
Accessory renal arteries	8/13 (62)
Endoleak at completion	
Type Ia	0 (0)
Type Ib	0 (0)
Type II	5 (14)
Type III	2 (5)
Complications	
Stroke	0 (0)
Myocardial infarction	0 (0)
Respiratory failure/pneumonia	0 (0)
Pancreatitis/any GI ischemia	0 (0)
Acute kidney injury	2 (5)
Access bleeding with surgical revision	3 (8)
Reintervention rate	2/37 (5)
ICU/IMC days	1 (0–8)
Hospitalization days	4.7 (2–17)

Data are presented as n (%) or median (range). ICU = Intensive care unit; IMC = Intermediate care unit, GI = gastrointestinal.

**Table 4 jcm-12-04708-t004:** Follow-up outcome of 37 patients with juxtarenal aortic aneurysms undergoing endovascular repair with fenestrated physician-modified endografts.

	All Patients (*n* = 37)
Follow-up, months	17.7 (1–47)
AAA diameter ^a^	
Increase ≥ 5 mm	1/36 (3)
Unchanged	25/36 (69)
Decrease ≥ 5 mm	10/36 (28)
Endoleaks ^a^	
Type Ia	0 (0)
Type Ib	0 (0)
Type Ic	0 (0)
Type II	12/36 (33)
Type III	0 (0)
Branch patency ^a^	
Occlusion	1/36 (3)
Stenosis	0 (0)
Device related outcomes ^a^	
Migration	0 (0)
Fracture	0 (0)
Graft limb occlusion	4/36 (11)
Days to secondary re-intervention	572 (4–1023)
Days to death	367 (74–1128)
All deaths	4 (11)
AAA related deaths	0 (0)

Data are presented as n (%) or median (range). ^a^ Excluding the patient with early open conversion.

## Data Availability

Data available on request due to restrictions e.g., privacy or ethical. The data presented in this study are available on request from the corresponding author.

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
