# Peer review of "Fenestrated Physician-Modified Endografts for Preservation of Main and Accessory Renal Arteries in Juxtarenal Aortic Aneurysms"

_jcm, 2023, doi:10.3390/jcm12144708_

Round 1

Reviewer 1 Report

Dear Authors,

Thank you for having submitted your evaluable paper.

You reported technical and clinical outcomes of PMEG in elective and urgent juxtarenal aneurysm. Rate of target vessel instability, in particular rate of endoleak, was very low for main renal arteries. Worse results were observed for accessory renal artery, in particular for those <4mm.

These are my concerns/suggestions:

1. As you report the use of this technique in urgent/emergent settings, please can you provide the mean duration of the complete procedure?

 2. Have you got any contraindications for the use of PMEG? In other words, are there any clinical or anatomical issues that you consider as predictor of failure?

3.  It would be interesting to know the mean length and oversize for the aortic endograft used in these patients.

4. How many fenestrations did you perform in this sub-group of patients? In other words, as the graft is completely deployed before target vessel cannulation, what are your technical tricks in case of malposition of the endografts?

5.      Many of your post-operative complications and reintervention are related to limb occlusion. Which was your post-operative anti-thrombotic therapy? After recanalization of such occluded limb, did you find the cause of occlusion?

Author Response

Reviewer 1

We thank you for the constructive review of this manuscript and for giving us the chance to submit a revised version. We have carefully considered your comments and have revised the manuscript accordingly. Additional data was collected where requested and all modifications are outlined below, and highlighted in the manuscript.  

  1. As you report the use of this technique in urgent/emergent settings, please can you provide the mean duration of the complete procedure?

We presented the total operation time in Table 3 with a median of 166 minutes. In urgent/emergent case, we work in two teams. One does the back-table modification and the other team prepares the percutaneous vessel access in both groins, including an occlusion balloon in the aorta in case of hemodynamic instability. For the three cases, the complete procedure lasted 105 minutes (1x fenestrated sm aortic cuff after secondary EVAR rupture due to type Ia endoleak), 180 minutes (3x fenestrated sm bifurcated graft for contained rupture) and 195 minutes (3x fenestrated sm bifurcated graft for 11cm symptomatic aneurysm). We added this information in Results page 6, line 163-168.

  1. Have you got any contraindications for the use of PMEG? In other words, are there any clinical or anatomical issues that you consider as predictor of failure?

Thank you for this interesting question. In general, any contraindication for a standard EVAR concerning the access vessel size, tortuosity, supra- and infrarenal angulation would be a contraindication for the PMEG. Severely calcified, angulated and narrow aortic necks could interfere with fenestration catheterisation. In this situation, we recommend partial deployment of the main body with upper extremity access for catheterisation of the fenestration through the bare metal crown. The only technical failure we had for the main renal arteries was in a severely calcified, angled and narrow aortic neck, which prevented the cannulation in first place. We added this in the discussion page 10, line 251-253.

  1. It would be interesting to know the mean length and oversize for the aortic endograft used in these patients.

We would like to thank the reviewer for pointing out the importance of the aforementioned parameters. We added the proximal oversizing for the endograft separately for cuffs and bifurcation endografts in Table 2. For cuffs we aim for 10% oversizing whereas for bifurcated grafts we go for 15-20% oversizing. The standard length used was 49mm for cuffs and 145mm or 166mm for bifurcated endografts. Leg lengths were adjusted accordingly intraoperatively. We added these information in the method section page 2, line 91-94.

  1. How many fenestrations did you perform in this sub-group of patients? In other words, as the graft is completely deployed before target vessel cannulation, what are your technical tricks in case of malposition of the endografts?

We created 47 fenestrations in 37 patients, which is now edited in Table 2. We leave the tip capture closed until at least one target vessel is cannulated. In case of three fenestrations we recommend a very slow and partial main body deployment which increases the margin for target vessel cannulation. In case of fenestration misalignement we use a ballon manoeuver to adjust the fenestration to the origin of the target vessel. We added this information in the discussion section page 10, line 267-270.

  1. Many of your post-operative complications and reintervention are related to limb occlusion. Which was your post-operative anti-thrombotic therapy? After recanalization of such occluded limb, did you find the cause of occlusion?

The patients were on unfraction heparin during hospitalisation and had double anti-plateled antiaggregation for 6 months (Aspirin and Clopidogrel). Patients previously set on (N)OAK had no anti-plated therapy added.

In two patients mechanical problems due to severe kinking or narrow native aortic bifurcation were detected. In these cases additional ballon stenting solved the mechanical problem and no sequential leg occlusion occured. One patient developed a local thrombosis with limb occlusion which was excluded with a stent graft. And in one patient a haematologic disorder with increased thrombogenicity was suspected as cause, without any signs of kinking or stenosis of the stent graft. This is mentioned in the results section, page 8, line 214-218.

Reviewer 2 Report

Dear authors,

many thanks for submitting your paper "Fenestrated Physician-modified Endografts for Preservation of Main and Accessory Renal Arteries in Juxtarenal Aortic Aneurysms" in the Journal of Clinical Medicine. This paper presents a single-centre experience of the PMEG technique for the treatment of JAAA. It is written in a very nice manner, however, some points remain that warrant author's attention. Major issues are labeled with asterix.

* How did you define JAAA?

- Can you provide a flowchart for all treated JAAA patients in your aortic centre? Do you use other techniques besides PMEG for complex AAA? What about open surgery? Please provide all of this information in one flowchart.

* Can you provide a rationale behind decission making process in your institution for the treatment of JAAA? I mean when do you decide to use some specific technique? This would be valuable for the readers to know.

- Can you provide in supplements what exactly data you collect?

* I miss some important CTA data such as alfa and beta proximal neck angle, thrombus and calcium presence (percentage), Iliac artery aneurysm, Iliac artery stenosis/tortuosity, etc.

* I miss also postoperative complications, such as MI, stroke, AKI, bleeding, etc.

- What happened with three patients who had unsuccessful ARA catheterization?

- Line 195, your median follow-up was 17.7mo, and you mention in this line 19mo. Please explain this.

* Please calculate the follow-up index.

Author Response

Reviewer 2

We thank you for the constructive review of this manuscript and for giving us the chance to submit a revised version. We have carefully considered your comments and have revised the manuscript accordingly. Additional data was collected where requested and all modifications are outlined below, and highlighted in the manuscript.  

* How did you define JAAA?

We used the same definition for juxtarenal aortic aneurysms as in the current ESVS guidelines (i.e., a short neck <10 mm). In case of accessory renal arteries the neck length was measured between the lowest accessory renal artery intented to preserve and the aortic aneurysm. For clarification we added this information in the methods section, page 2, line 65-67.

Can you provide a flowchart for all treated JAAA patients in your aortic centre? Do you use other techniques besides PMEG for complex AAA? What about open surgery? Please provide all of this information in one flowchart.

We made the changes accordingly, and added this flowchart per request in figure S1 in the supplementary appendix and mentioned it in the method section page 2, line 75-76.

* Can you provide a rationale behind decission making process in your institution for the treatment of JAAA? I mean when do you decide to use some specific technique? This would be valuable for the readers to know.

We created a flowchart with the rationale behind treatment decision making process in patients with juxtarenal AAA in our aortic centre and added this information in figure S2 which is mentioned in the methods section, page 2, line 76-77.

- Can you provide in supplements what exactly data you collect?

We made the changes accordingly, and added the requested data in Table S1, which is mentioned in the methods section, page 2, line 66.

* I miss some important CTA data such as alfa and beta proximal neck angle, thrombus and calcium presence (percentage), Iliac artery aneurysm, Iliac artery stenosis/tortuosity, etc.

We would like to thank the reviewer for pointing out the importance of the aforementioned parameters. These parameters are now added in Table 2.

* I miss also postoperative complications, such as MI, stroke, AKI, bleeding, etc.

Thank you for this comment, we added this now in Table 3.

- What happened with three patients who had unsuccessful ARA catheterization?

These accessory renal arteries remained patent throughout the follow-up period and there was no endoleaks detected due to alignment of the fenestrations with the target vessel. We added this figure in supplements and added it in the results section page 6 and 9, line 191 and 228.

- Line 195, your median follow-up was 17.7mo, and you mention in this line 19mo. Please explain this

We thank the reviewer for pointing out this error, we made the changes accordingly, the median follow-up was 17.7 months (range 1-47).

* Please calculate the follow-up index

We thank the reviewer for this suggestion, and we therefore made the calculations accordingly. Life status was assured at the study closure date in all patients. We added this information in the methods section, page 2, line 65-66 and the follow-up index was added in the results section, page 8, line 209-201.

Round 2

Reviewer 2 Report

Dear authors,

many thanks for resubmitting your paper at the JCM. You have put a significant effort into the first revision process and the overall quality and clarity of the manuscript have improved a lot. 

I have the following comment:

- Figure S2 is not fully clear to me. Can you please elaborate further on what is the decision process? You can describe this in the figure legend.

- Do you have data about the renal function (creatinine and GFR) at the admission, after the discharge, and after the median follow-up period? It would be nice to have this since we are speaking about the JAAA. Also, making a univariate analysis and identifying predictors of short- and long-term renal function deterioration after complex endovascular repair would be great to have. I suppose you used RIFLE/KDIGO criteria to define acute kidney injury (you should mention this in the methodology). For long-term renal function impairment, you can use the definition drop of more than 20% of GFR compared to the baseline values.

Author Response

Dear Reviewer

Please find attached our reply to your valuable comments.
